# SPATIO-TEMPORAL DISENTANGLED REPRESENTATION LEARNING FOR MOBILITY FORECASTING

## ABSTRACT

Spatio-temporal (ST) prediction task like mobility forecasting is of great significance to traffic management and public safety. There is an increasing number of works proposed for mobility forecasting problems recently, and they typically focus on better extraction of the features from the spatial and temporal domains. Although prior works show promising results on more accurate predictions, they still suffer in characterising and separating the dynamic and static components, making it difficult to make further improvements. Disentangled representation learning separates the learnt latent representation into independent variables associated with semantic factors (Duan et al., 2019). It offers a better separation of the spatial and temporal features, which could improve the performance of mobility forecasting models. In this work, we propose a VAE-based architecture for learning the disentangled representation from real spatio-temporal data for mobility forecasting. Our deep generative model learns a latent representation that (i) separates the temporal dynamics of the data from the spatially varying component and generates effective reconstructions; (ii) is able to achieve state-of-the-art performance across multiple spatio-temporal datasets. Moreover, we investigate the effectiveness of our method by eliminating the non-informative features from the learnt representations, and the results show that models can benefit from this operation.

## 1 INTRODUCTION

Spatio-temporal prediction tasks like mobility forecasting are critically important for smart city applications (Zheng et al., 2014). With the help of the rapid deployment of IoT (Internet of Things) devices and sensors, massive amounts of saturated ST datasets are available, and many researchers have put their efforts to improve the performance of ST prediction (Zhang et al., 2016; Wang et al., 2018; Zonoozi et al., 2018; Jin et al., 2018; Yao et al., 2019b). A central problem in deep learning for crowd flow prediction tasks is the extraction of features from both spatial and temporal domains. Researchers first tried to extract the entangled spatio-temporal features directly from the data (Zhang et al., 2016). Then, they tried to extract features from spatial and temporal domains separately. Wang et al. (2018) and Zonoozi et al. (2018) proposed models that extract periodic representations or short-term temporal features directly from the ST data using recurrent-based convolution operations and find their effectiveness in producing more accurate predictions. Based on that, in STDN (Yao et al., 2019b), the explicit extraction of the long-term periodic information was shown to also improve the prediction.

Although the existing approaches appear to be powerful in terms of results and predictions, two major challenges hinder these models: **1) Difficulty in characterising dynamic and spatial components.** Spatio-temporal correlation is more complex since it comprises dependencies from both the spatial and temporal regions. Models like LDRSN (Tian et al., 2020) and RegionTrans (Wang et al., 2018) are capturing temporal dependencies explicitly. However, their complicated structure makes it hard to characterise and validate the effectiveness of their dynamic and spatial components. **2) Difficulty in separating the extraction of spatial and temporal features.** Except for the spatial and temporal features from the most recent data, many approaches try to extract long-term/periodical features to improve the performance. However, without an explicit separation mechanism, the extracted long-term temporal dependencies introduce irrelevant noises to the predictor (factors do not vary with time and are only relevant to the long-term sequence). The extent to how generative models can model, extract and disentangle the spatial and temporal features in ST-raster data is an open problem.

To address the above challenges, we introduce the disentangled representation learning to the mobility forecasting task. Disentangled representation learning, which separates the learnt representation into independent variables such that each variable relates to one semantic factor of sensory data (Bengio et al., 2013), offers a solution for the problem mentioned above. For a spatio-temporal task like mobility forecasting, an ideal disentangled representation should have the ability to separate time-relevant components from the factors that don't vary with time, which can help improve the predictor's performance.

Many prior works have explored disentangled representation learning for spatio-temporal data (Hsu et al., 2017; Li & Mandt, 2018; Denton & Birodkar, 2017; Zhu et al., 2020). Most of them assume that the features can be disentangled into two sets: a set of dynamic features that extract the temporal correlations and a set of static features that describe factors that are constant through the input sequence. However, since most of them focus on the movement of some predefined objects, their assumption is inaccurate when applying to datasets for mobility forecasting. For example, a sudden car accident might show no relationship to the temporal influence from the previous timestep and the static traffic network structure. Therefore, there are still gaps in how to model the spatio-temporal data using the disentangled representation learning method. To address this problem, we assume that each timestep has its own temporal features and features that do not vary with time. By doing so, our model can better formulate the spatio-temporal data, and we can achieve controlled data generation frame by frame.

In this work, we proposed a VAE-based model to learn disentangled spatio-temporal representation. Compared to the conventional methods for the ST prediction problem, our approach extracts the entangled features first and then explicitly separates them into temporal variables and spatial variables using disentangled representation learning method. This will force the model to keep the learnt spatial/temporal features as mutually exclusive as possible. For applying disentangled representation learning method to real ST data, we assume that each frame has its own spatial and temporal variables and separate these two groups with auxiliary regularisation. It helps the model to formulate the complicated spatio-temporal sequence. Our experimental results (see Section 4) show that the learnt representations have a similar level of performance with the current state-of-the-art methods. Our key contributions can be summarised as follows:

1. We propose a novel approach to learn disentangled spatio-temporal representations for mobility forecasting tasks. The learnt representation is separated into two independent groups: spatial and temporal factors.

2. We conducted several experiments on multiple spatio-temporal datasets and used the learnt representation for mobility forecasting. Results show that our methods achieve state-of-the-art performance compared to other baseline mobility forecasting methods.

3. We investigate the effectiveness of our methods under the "Closeness, Period, Trend" scheme and how to further improves the model's performance by selecting the informative features from the learnt representations.

## 2    RELATED WORK

**Deep spatial-temporal networks for mobility forecasting:** In order to make accurate traffic prediction, many researchers have paid attention to capturing the spatio-temporal dependencies hidden behind the traffic data. Besides the conventional methods like Seasonal ARIMA (Moreira-Matias et al., 2013), deep learning methods are increasingly used in more and more works for mobility forecasting.

ST-ResNet (Zhang et al., 2016) was proposed to capture the spatial dependencies through a stack of residual convolution layers. It also stacks the frame from a near and a distant time period to capture temporal dynamics. The goal of using the residual units is to overcome the gradient vanishing problem, and the results show that capturing distant spatial features can improve the performance of mobility forecasting. DeepSTN+ (Lin et al., 2019) proposed a ResPlus unit that is capable of capturing long-range spatial correlations. Although they succeed in extracting distant spatial features, they lack attention on capturing the temporal dependencies. To further explore the effectiveness of temporal features, recurrent-based approaches were introduced to capture the temporal correlations. PCRN (Zonoozi et al., 2018) was proposed, which first extract entangled spatio-temporal representation

using a convolutional recurrent network (CRN) and then updating the periodic representations by CRN's hidden state. Attention-based LSTM methods are adopted by STDN (Yao et al., 2019b) and ST-DCCNAL (Li et al., 2019), which try to capture long-term temporal dependencies. In summary, our proposed method differs from other methods in the explicit separation of spatial and temporal features using a disentangled representation mechanism.

**Disentangled Representation Learning for Spatio-temporal data:** Most prior works on the disentangled representation learning problem are developed based on Variational Autoencoders (VAE) (Kingma & Welling, 2013), which is an unsupervised generative learning method. $\beta$-VAE, proposed by Higgins et al. (2016), forces the inference model to disentangle the latent representation by adding a new hyperparameter $\beta$ to create an information bottleneck on the prior. FactorVAE (Kim & Mnih, 2018) further breaking down the objective function and try to enhance disentanglement by penalising the total correlation of the learnt representation. As for sequence modelling, a number of prior publications have extended VAE to video and speech data (Fabius & Van Amersfoort, 2014; Chung et al., 2015; Bayer & Osendorfer, 2014). These models, although being able to generate realistic sequences, do not explicitly disentangle the representation of time-invariant and time-dependent information. Thus it is inconvenient for these models to perform tasks such as controlled generation. S3VAE (Zhu et al., 2020) is proposed to separate static and dynamic factors of sequential data. Another approach proposed by (Li & Mandt, 2018) is also focusing on separating the dynamic factors from static factors. Although they share a similar idea which uses an RNN-based architecture to extract dynamic factors for each timestep, they use different prior setups. Each frame in (Li & Mandt, 2018) has its own content features while the time-invariant variables are shared by the whole sequence in S3VAE. As for the spatial-temporal type of data like video, SV2P (Babaeizadeh et al., 2017) uses the variational model to extract the time-invariant latent and make predictions for multiple frames. Models like (Denton & Birodkar, 2017; Hsieh et al., 2018) try to factorise each frame into a stationary part and a temporally dynamic component.

## 3 SPATIO-TEMPORAL VAE MODEL

### 3.1 SPATIO-TEMPORAL DATA

In this work, we will use $\mathcal{D} = \{X^i\}^{i=1:N}$ to denote a spatio-temporal raster dataset that comprises $N$ i.i.d. sequences. Each $X \equiv x_{1:T} = \{x_1, x_2, ..., x_T\}$ in that dataset denotes a sequence of raster data with $T$ frames. Since we focus on mobility forecasting tasks using grid-based spatial representations, which all have the similar dimensions $H \times W$. Hence each frame $x_t \in \mathbb{R}^{H \times W}$ represents the mobility flow of a certain area at a given time interval $t$.

### 3.2 SEPARATING SPATIAL FEATURES FROM TEMPORAL FEATURES

Many prior works have explored disentangled representation learning for spatio-temporal data. Models like FHVAE (Hsu et al., 2017) tried to separate global variables from segment (dynamic) variables for speech data. DSVAE (Li & Mandt, 2018), DRNET (Denton & Birodkar, 2017) and S3VAE (Zhu et al., 2020) aim at the disentangled representation for video, which also factorised latent variables into static and dynamic parts. However, current approaches for learning the disentangled representation on sequence or spatio-temporal data like video assume that there is a fixed content or object shared by all frames in the sequence since they are using video datasets like the Stochastic Moving MNIST (Denton & Fergus, 2018) and Sprite (Li & Mandt, 2018). Sequences in these datasets often comprise images describing the movement of a set of the same numbers or virtual avatars. Under this context, static features describing this object can be extracted, and its performance can be evaluated by the accuracy of a classification task. However, for most of the real spatio-temporal datasets, the static features are not enough. For example, in a traffic flow dataset, in addition to the fixed structure of the traffic network and the temporal influences, there might be some hotspot suddenly emerges. At that specific time, the rise of those events shows no correlation to the temporal features and the fixed content. Therefore, we assume that for each frame, data is generated based on two sets of features: a set of temporal features which comprise the influence from the sequence before it and a set of spatial features that describe the structure of the network and local events. In this work, we propose a novel architecture that extracts mixed (entangled) feature maps for each timestep in the input sequence and then separates them into temporal and spatial variables. It allows us to analyse and controlled generate each frame separately.

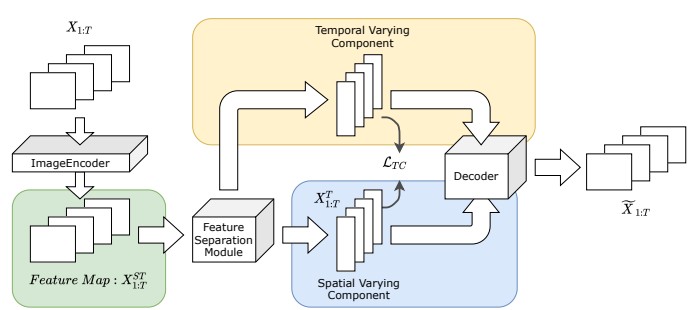 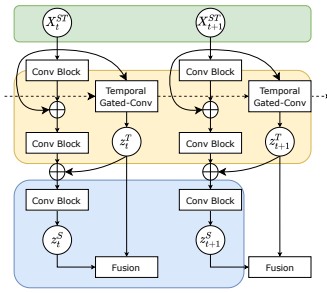

(a) The framework of our proposed model

(b) A detailed structure of the Feature Separation module

Figure 1: The framework of our proposed model. Each frame of a spatio-temporal sequence $x_{1:T}$ is fed into the Image Encoder first to extract entangled high-level feature maps, which are then passed through a temporal gated convolution layer to capture the time-varying variables (temporal features) $z_{1:T}^{Te}$. Then the time-irrelevant variables (spatial features) $z_t^{Sp}$ for each frame $x_t$ is captured through several convolutional blocks. In addition to the objective of the VAE, a Total Correlation regularizer is used to encourage the disentanglement of the learnt representation. Reconstruction input is generated for each frame based on the sampled latent from both domains.

The architecture of our method for spatio-temporal disentangled representation learning is illustrated in Fig.1a. It consists of three major modules: (i) *Image Encoder* to yield initial feature maps from the input spatio-temporal sequence. (ii) A *Feature Separation* module to separate the spatial and temporal variables from the initially mixed feature maps. (iii) A *Decoder* to reconstruct the sequence frame by frame based on their corresponding spatial and temporal variables.

The Image Encoder comprises several layers of CNN, and the feature maps are extracted from each frame separately. The goal here is to yield some higher-level features for the next step. An Instance Normalisation layer is used after each convolutional layer to improve the reconstruction results since using batch normalisation might remove the instance-specific contrast information from the data, which is useful in the later step (Ulyanov et al., 2016).

Intuitively, without any disentanglement constraints, the feature maps that come out of the Image Encoder should be the entanglement of features from both spatial and temporal domain. Therefore, the goal for the Feature Separation module is to separate them explicitly. For temporal features, inspired by (Yu et al., 2017; Gehring et al., 2017), we use a Gated Convolutional structure to capture temporal dynamics from the mixed feature maps. This temporal gated convolution layer contains two 1-D causal convolution layers. As Fig.1b shows, those 1-D convolution operations will be applied on the time axis for each pixel of the feature maps, which means the capture of the temporal dynamics from each mixed high-level feature. It is much easier to understand the usage of the Image Encoder here since the 1-D convolutional operation on the high-level features map is definitely better for separating the temporal dynamics than applying them on the individual pixel of the original input.

The Decoder module is used to reconstruct each frame $x_t$ by using their corresponding spatial variables $z_t^{Sp}$ and temporal variables $z_{1:t}^{Te}$ as input.

### 3.3 SPATIO-TEMPORAL VAE MODEL

**Priors:** In this work, we proposed a spatio-temporal variational autoencoder architecture to learn disentangled representations from ST raster datasets. Our assumption for variational autoencoder is that each input $x_t$ is generated from a corresponding latent representation $z_t$ which can be separated into two disentangled subgroup: variables $z_t^{Sp}$ which contains the spatial (time-irrelevant) features and the temporal (time-varying) features $z_{1:t}^{Te}$.

On the one hand, since the spatial variables $z_t^{Sp}$ is considered time-irrelevant, therefore its prior is defined as a standard Gaussian distribution $z_t^{Sp} \sim \mathcal{N}(0,1)$ and the spatial variables for the whole sequence can be formed as $z_{1:T}^{Sp} = \prod_{t=1}^{T} z_t^{Sp}$. On the other hand, the prior of the temporal dynamic

variables follow a sequential prior $z_{1:t}^{Te} = z_t^{Te}|z_{<t}^{Te}$. The distribution of this sequential prior is defined as $z_t^{Te}|z_{<t}^{Te} \sim \mathcal{N}\left(\mu_t, diag\left(\sigma_t^2\right)\right)$ where $\left[\mu_t, \sigma_t^2\right] = \phi^{Te}\left(z_{<t}^{Te}\right)$. The mean $\mu_t$ and variance $\sigma_t^2$ are parameters that are conditioned on all of its previous temporal variables $z_{<t}^{Te}$ and can be parameterised by the gated-convolution layer. Thelatent prior $z_t$ that combines both the spatial and temporal variables is formed as:

$$p(z_t) = p(z_t^{Sp})p(z_{1:t}^{Te}) = p(z_t^{Sp}) \prod_{t=1}^{T} p(z_t^{Te}|z_{<t}^{Te}) \tag{1}$$

**Generative model:** For the generative model, we assume that the generation of each frame $x_t$ at a given time $t$ depends on the combination of its corresponding spatial variables $z_t^{Sp}$ and temporal variables $z_t^{Te}$. Therefore, the generation process for the single frame $x_t$ in the whole sequence $x_{1:T}$ can be formed as:

$$p_\theta(x_t, z_t) = p_\theta(x_t|z_t)p(z_t) = p_\theta(x_t|z_t)p\left(z_t^{Sp}\right) \prod_{t=1}^{T} p\left(z_t^{Te}|z_{<t}^{Te}\right) \tag{2}$$

where $\theta$ are the parameters for the decoder.

**Inference models:** We use a deep structured model as an encoder to approximate the posterior distribution, which can factorise the latent z into disentangled spatial and temporal components. The amortised variational distribution is formed as:

$$q_\phi\left(z_{1:T}^{Te}, z_T^{Sp}|x_{1:T}\right) = \prod_{t=1}^{T} q_\phi\left(z_t^{Sp}|x_t\right) \prod_{t=1}^{T} q_\phi\left(z_t^{Te}|x_{<t}\right) \tag{3}$$

## 3.4 Loss

In this work, the objective of our proposed method is defined as the combination of the VAE loss and total correlation regularisation. It can be formulated as: $\mathcal{L} = \mathcal{L}_{VAE} + \beta\mathcal{L}_{TC}$ where $\beta$ is the hyperparameter for the TC regularisation. Theoretically, applying higher value on $\beta$ will emphasise the disentanglement of the learnt representation and lead to better separation of the spatial and temporal variables. We estimate the objective based on FactorVAE's approach (Kim & Mnih, 2018).

**VAE Objective Function:** The objective function of the our method is derived from the variational lower bound (Evidence Lower Bound, ELBO) of the vanilla VAE (Kingma & Welling, 2013) and is formed as follow:

$$\mathcal{L}_{VAE}(\theta, \phi; x_{1:T}) = E_{q_\phi(z_{1:T}^{Sp}, z_{1:T}^{Te}|x_{1:T})}[\sum_{t=1}^{T} log\, p_\theta(x_t|z_t^{Sp}, z_t^{Te})] - \sum_{t=1}^{T} D_{KL}(q_\phi(z_t^{Sp}|x_t)||p_\theta(z_t^{Sp}))$$

$$- \sum_{t=1}^{T} D_{KL}(q_\phi(z_t^{Te}|x_{\leq t})||p_\theta(z_t^{Te}|z_{<t}^{Te})) \tag{4}$$

Note that S3VAE (Zhu et al., 2020) already propose a sequential VAE that consider the continuity of dynamic variables, but it only assumes that there are a whole set of statice features share by the whole sequence. In contrast, we model the temporal and spatial features for each time step independently, resulting in detailed information for emergencies in mobility forecasting data.

**Total Correlation Regularization:** To encourage the overall disentanglement of the learnt representation, we introduced the *Total Correlation (TC)* among variables as a regularization term. It quantifies the dependency among a set variables (Alfonso et al., 2010). Experimental results from $\beta$-TCVAE (Chen et al., 2018) and FactorVAE (Kim & Mnih, 2018) show that, by amplifying the penalty on this term, the dependence between the variables is reduced hence emphasising the disentanglement.

In this work, we estimate the total correlation using the same approach like FactorVAE (Kim & Mnih, 2018), i.e. by introducing a discriminator and using the independence testing trick and the density-ratio trick to approximate the KL term in the above equation.

## 4 EXPERIMENTS

### 4.1 DATASET AND METRICS

In this work, we focus on learning disentangled representation of spatio-temporal raster data. Therefore, we choose to conduct the experiments on the following three real-world urban flow datasets:

**BikeNYC** (Lin et al., 2019) is a bike usage data collected from New York City's Citi Bike bicycle sharing service, which records the trajectory of all shared bikes in the system. This work covers the time period from 2014-04-01 to 2014-09-30.

**TaxiNYC** (Yao et al., 2019a) is a dataset that contains taxi in-out flow data taxi New York City, created from the NYC-Taxi GPS data, which covers the period from 2015-01-01 to 2015-03-01.

**TaxiBJ** (Zhang et al., 2016) comprises the taxi in-out flow data that aggregate the taxi GPS position in Beijing from 2013 to the year 2016. Although it span across 4 consecutive years, the data is not continuous (covers the period: 2013-07-01 to 2013-10-30, 2014-05-01 to 2014-06-30, 2015-03-01 to 2015-06-30, and 2015-11-01 to 2016-04-10).

For disentangled representation learning, we use all of the data to train the feature separation module, and for the mobility forecasting tasks, we use the same setup described in (Xue & Salim, 2021), i.e., the first 80% data is used for training the prediction model, and the rest 20% data is used for testing. Besides, Min-Max normalization is adopted to transform the urban flow values into the range [0, 1] for better training purpose. We evaluate the effectiveness of our models with two commonly used metrics: the Mean Absolute Error (MAE) and the Root Mean Square Error (RMSE).

### 4.2 IMPLEMENTATION DETAILS

To compare the performance of our model in mobility forecasting with the current state-of-the-art methods, we use the "Closeness, Period, Trend" scheme to form the input of our prediction model. It is widely used in the urban flow prediction area, where all three of them comprise a sequence of raster data. It designs a set of unique input sequences, namely Closeness, Period, and Trend, which correspond to the recent time intervals, daily periodicity, and weekly trend, respectively (Zhang et al., 2016). Those three sequences are then fed as the input of their models.

As shown in Figure 2, in this work, we use the same setup used by VLUC-Net (Jiang et al., 2019), where the Closeness sequence contains the previous six steps before the prediction target; the Period is the sequence of the previous day, and the Trend comprises data from the pre-

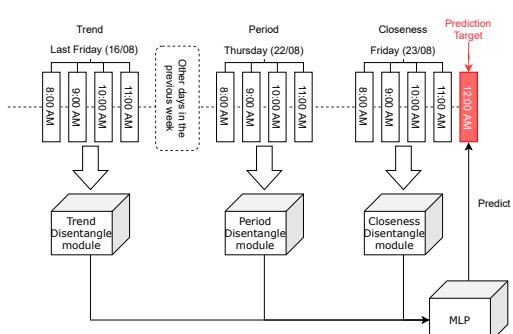

Figure 2: Illustration of the "Closeness, Period, Trend" components and the actual architecture in the mobility forecasting experiment.

vious week. We first train our model to learn disentangled representation on the closeness, period and trend sequences separately. Then the learnt representations are fused to train a Multi-Layer perceptron (MLP) regressor to predict the next frame of the sequence. It should be noted that the input of the MLP is not the actual features $z$, but the learnt $\mu$ and $\sigma$ of the distribution. The logic of doing this is because the sampling part will introduce uncertainty which has a huge impact on the training.

Table 1: Effectiveness Evaluation of Traffic In-Out Flow Prediction. Lower is better.

| | BikeNYC | | TaxiNYC | | TaxiBJ | |
|---|---|---|---|---|---|---|
| | RMSE | MAE | RMSE | MAE | RMSE | MAE |
| HA | 4.874 | 1.500 | 21.535 | 7.121 | 45.004 | 24.475 |
| CNN | 4.511 | 1.574 | 16.741 | 6.884 | 23.550 | 13.797 |
| ConvLSTM | 3.174 | 1.133 | 12.143 | 4.811 | 19.247 | 10.816 |
| ST-ResNet(Zhang et al., 2016) | 3.191 | 1.169 | 11.553 | 4.535 | 18.702 | 10.493 |
| DMVST-Net(Yao et al., 2018) | 3.521 | 1.287 | 13.605 | 4.928 | 20.389 | 11.832 |
| DeepSTM+(Lin et al., 2019) | 3.205 | 1.245 | 11.420 | 4.441 | 18.141 | 10.126 |
| STDN(Yao et al., 2019a) | 3.004 | 1.167 | 11.252 | 4.474 | **17.826** | **9.901** |
| VLUC-Net(Jiang et al., 2019) | 3.119 | 1.124 | **10.654** | 4.157 | 18.378 | 10.325 |
| Without explicit disentanglement | 5.412 | 1.537 | 19.985 | 6.281 | 21.631 | 12.202 |
| Spatial Only | 3.584 | 1.285 | 14.640 | 4.812 | 21.111 | 12.435 |
| Temporal Only | 3.107 | 1.167 | 14.175 | 4.669 | 19.825 | 11.638 |
| Ours | **2.903** | **1.119** | 12.022 | **4.055** | 19.185 | 10.741 |

## 4.3 MOBILITY FORECASTING: COMPARISON AGAINST OTHER METHODS

In our experiments, we compare our method against the following mobility flow prediction methods: Historical Average (HA); Convolutional Neural Network(CNN); Convolutional LSTM (ConvLSTM); ST-ResNet(Zhang et al., 2016); DMVST(Yao et al., 2018); DeepSTN+(Lin et al., 2019); STDN(Yao et al., 2019a); VLUC-Net(Jiang et al., 2019).

The overall evaluation results on effectiveness are summarised in Table 1 for TaxiBJ, TaxiNYC and BikeNYC. The upper half of Table 1 shows the results of baseline methods on those datasets, and the lower part shows the results of our proposed approach. The best result for each column is given in bold, and methods except HA, CNN, and ConvLSTM are all the current state-of-the-art methods in crowd flow prediction. In general, we can find that our method shows the best results on the BikeNYC and TaxiNYC datasets and compatible performance on the TaxiBJ dataset.

Besides the crowd flow prediction with a full feature set, we also trained models with only spatial/temporal features to see which set contributes more to the mobility forecasting. We find that the prediction performance of the models trained on the temporal feature set leads those trained on the spatial features by a relatively large margin. This shows that for the crowd flow prediction task, the extraction of the temporal dependencies is important than the spatial dependencies, which also coincides with the research direction in this area. It should also be noted that the models trained on the temporal feature already show compatible results with the current state-of-the-art methods. The is still room for improvement since the input of our method is the raw data alone with no context information.

## 4.4 ABLATION STUDY

To explore the effectiveness of each module in the "Closeness, Period, Trend" scheme, we perform an ablation study consider different configurations of the input sequences. The detailed configuration and their corresponding results are summarised in Table 2. The left half of the table shows the sequences used for each setup. By discarding extracted features from some sequences in the scheme, we are able to evaluate their effectiveness in regrading the mobility forecasting task. Since the quality of the learnt disentangled representation is sensitive to the choice of hyperparameters (Duan et al., 2019), we trained multiple feature separation modules with different hyperparameters. The mean and standard deviation of each configuration on all three datasets are presented in the right half of Table 2.

We can first find that the performance of models using period or trend sequence alone is poor, and sometimes adding those extra information does not help the mobility forecasting task. On average, the configuration that achieves the best results for all three datasets is not configuration C6. Although the features from the trend sequence improve the results for the BikeNYC dataset by a tiny margin, introducing the middle/long-term data worsens the performance for both the TaxiNYC and TaxiBJ

Table 2: Seven different configurations of the input sequences and their corresponding RMSE and MAE results. It should be noted that C6 is the widely-used setup of the current state-of-the-art mobility forecasting models

|  | Configuration | | | BikeNYC | | TaxiNYC | | TaxiBJ | |
|---|---|---|---|---|---|---|---|---|---|
|  | Closeness | Period | Trend | RMSE | MAE | RMSE | MAE | RMSE | MAE |
| C0 | ✓ | ✗ | ✗ | 3.660 ±1.88 | 1.209 ±0.45 | 14.323 ±1.31 | **4.670** ±**0.35** | 22.035 ±6.87 | **14.078** ±**3.57** |
| C1 | ✗ | ✓ | ✗ | 5.924 ±2.12 | 1.658 ±0.61 | 25.364 ±2.06 | 7.268 ±0.45 | 35.222 ±3.72 | 21.785 ±2.24 |
| C2 | ✗ | ✗ | ✓ | 4.217 ±2.17 | 1.361 ±0.53 | 22.387 ±2.58 | 6.841 ±0.70 | 42.361 ±2.93 | 25.967 ±1.85 |
| C3 | ✓ | ✓ | ✗ | 3.707 ±2.43 | 1.222 ±0.54 | 14.431 ±1.60 | 4.704 ±0.44 | 22.140 ±6.76 | 14.182 ±3.52 |
| C4 | ✓ | ✗ | ✓ | 3.558 ±1.77 | **1.198** ±**0.44** | 17.020 ±2.38 | 5.281 ±0.56 | 22.585 ±7.21 | 14.434 ±3.66 |
| C5 | ✗ | ✓ | ✓ | 4.406 ±1.92 | 1.388 ±0.49 | 22.634 ±2.34 | 6.692 ±0.54 | 36.454 ±3.24 | 22.492 ±1.92 |
| C6 | ✓ | ✓ | ✓ | 3.601 ±2.10 | 1.210 ±0.50 | 16.677 ±2.29 | 5.173 ±0.52 | 22.504 ±6.97 | 14.388 ±3.52 |

datasets. One of the reasons might relate to the size of the representation. Although there might be some information that can contribute to the prediction, using all three sequences tripling the dimension of the input and let the noise offset the potential benefit. Besides, we can find out that the variance results are high for all three datasets, which agree with the assumption that the quality of the learnt representation regarding the mobility forecasting task is sensitive to the choice of hyperparameters.

## 4.5 INFORMATIVE FEATURES

Given the thought that the dimension of MLP's input might be too large and the learnt representations from the previous section, we want to investigate whether we can find the "informative" features from the learnt representations. And for those "normal" representations (contains valuable information but does not achieve the best results), will the models benefit when we exclude those "non-informative" features from their input. In our work, we define the "informative" features using the definition from Unsupervised Disentanglement Ranking (UDR)(Duan et al., 2019): A latent dimension is treated as an "informative" feature if it learns a latent posterior that diverges from the prior.

$$I_{KL}(a) = \begin{cases} 1 & KL(q_\phi(z_a|x)||p(z_a)) > 0.001 \\ 0 & otherwise \end{cases} \tag{5}$$

where $a$ is the index of the variable. We use a smaller threshold here since the number of features is larger than the UDR (Duan et al., 2019). To avoid the influence from random noise, we test the performance of mobility forecasting on all representations that we get from the previous section, and the results are summarised in Figure 3a.

First, it should be noted that there exist representations that do not contain any informative features. As shown in the figure, those "poorly-learnt" representations perform a lot worse compared to the ones with informative features. That proves that a "well-disentangled, well-learnt" representation can contribute to the downstream task, at least for the mobility forecasting task. To prove the necessity of the filtering operation (remove the "non-informative" features), we first decrease the representation size and train the feature separation module to extract representations. The results show that when the size is below a certain threshold, all representations with smaller sizes will not contain any informative features. And the ones containing informative features will always have some space. That indicates the necessity of the filtering operation since we still haven't found a solution to learn disentangled representation that only contains informative features.

Another thing that we want to investigate is whether the models can benefit when removing those non-informative features from their input. Therefore, we train those MLPs with only the informative

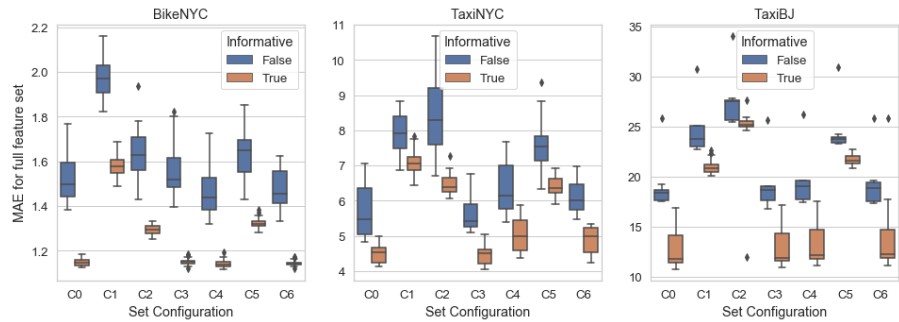

(a) Results of different configurations for representation contain/not contain informative features.

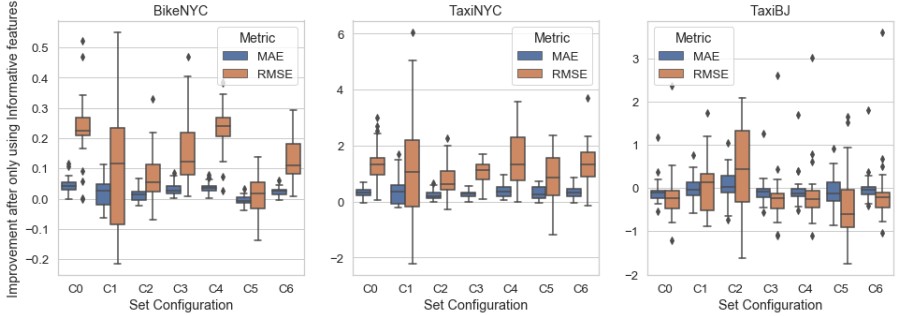

(b) Improvements of different configurations after only using informative features as input.

Figure 3: The boxplot of RMSE and MAE results for different representations (learnt with different hyperparameters).

features and calculate the difference in performance and summarised the results in Figure 3b. It is worth noted that the MLPs have the same structure except the first layer due to the reducing amount of input features.

In summary, for the BikeNYC and TaxiNYC datasets, we can find out that, although the MLP that trained with the best representation does not get performance improvement, the majority of the MLPs that trained with "normal" representations show better results after removing the "non-informative" features. As for the TaxiBJ dataset, there is a slight drop in performance but only with a tiny margin. The reason for that might be due to the different sizes and data distribution of TaxiBJ. The overall grid size for TaxiBJ is larger than the other two datasets, and the Cumulative Distribution Function (CDF) shows that it contains more large values in the dataset, which might lead to different behaviour when we are removing the "non-informative" features. More details can be found in the Appendix.

## 5    CONCLUSION

Spatio-temporal (ST) prediction tasks like mobility forecasting have attracted significant attention since they greatly influence traffic management and public safety. We introduce the disentangled representation learning method and modify it to fit spatio-temporal data. The experimental evaluation results show that our method can achieve state-of-the-art performance and is able to extract desirable spatial/temporal features. Moreover, we investigate the effectiveness of recent/middle/long-term temporal features and find that sometimes our method can achieve state-of-the-art results without long-term temporal features. Finally, we also demonstrated that a well-learnt representation shows better results in mobility forecasting tasks and removing the non-informative features from the input of downstream models sometimes can also boost performance. Hence, we hope that our method can contribute to the mobility forecasting task by introducing the disentangled representation learning mechanism. One future direction of this work is forcing the model to learn a more compact representation that only contains "informative" features. A better mechanism to separate the spatial/temporal features and link them to real semantics is also needed to be investigated.

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

# A   APPENDIX

## A.1   THE CUMULATIVE DISTRIBUTION FUNCTION (CDF) OF THE DATASETS

The Cumulative Distribution Function (CDF) of all three datasets is shown in Figure 4. We can find that the TaxiBJ dataset contains more data with larger values, therefore shows a different data distribution comparing to the other two datasets.

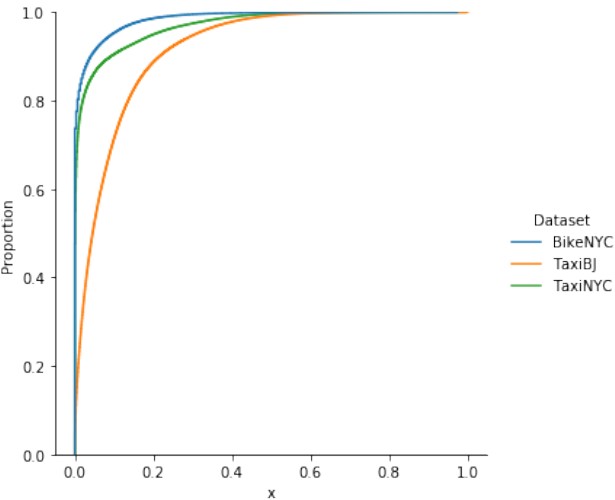

Figure 4: The Cumulative Distribution Function of all three datasets.

## A.2   VAE LOSS DECOMPOSITION AND TOTAL CORRELATION REGULARISER

Variational Autoencoder (VAE) is a generative model that tries to approximate a posterior distribution $p_\theta(z|x)$ by a neural network component $q_\phi(z|x)$. The generative part means that in this model, the data $x$ is considered generated by some random process modelled as $p_\theta(z)p_\theta(x|z)$. And we optimize $\theta$ to $\theta^*$ by maximizing the marginal likelihood:

$$log\, p_\theta(x) = D_{KL}(q_\phi(z|x)||p_\theta(z|x)) + \mathcal{L}_{VAE}(\theta, \phi; x) \tag{6}$$

Due to the intractability of the posterior distribution, it is not likely to calculate the marginal likelihood directly. However, the author provided a variational lower bound (Evidence Lower Bound, ELBO), which can be optimized using stochastic gradient descent:

$$\mathcal{L}(\theta, \phi; x) = -D_{KL}(q_\phi(z|x)||p_\theta(x)) + E_{q_\phi(z|x)}[log\, p_\theta(x|z)] \tag{7}$$

The second part in Eq. 7 represents the *reconstruction loss*, and by minimising this term, it forces the model to generate reliable reconstructions of the input and the synthesis data with better quality. The first term is a regularisation term which can be further tear down for the purpose of better disentanglement on the latent representation.

$\beta$-VAE (Higgins et al., 2016) is the first model for introducing the disentanglement. By adding a hyper-parameter $\beta$ for penalising the Kullback-Liebler divergence term harder, the representation tends to become disentangled, and the reason behind it is quite straight forward if we decompose that term into the following form (Kim & Mnih, 2018):

$$D_{KL}(q_\phi(z|x)||p_\theta(z|x)) = I(x,z) + D_{KL}(q_\phi(z)||p(z)) \tag{8}$$

The first term describes the *mutual information* between the input and its latent representation while the second term pushes the distribution of the latent towards the Gaussian prior thus emerging the

disentanglement. Although the hyperparameter $\beta$ emphasises, it suppresses the model's ability to produce a high-quality reconstruction. To address that limitation, the disentanglement term in Eq. 8 is further decomposed into the following form(Chen et al., 2018):

$$D_{KL}(q_\phi(z)||p(z)) = D_{KL}\left(q(z)||\prod_{j=1} q(z_j)\right) + \sum_d D_{KL}(q_\phi(z_j)||p(z_j)) \tag{9}$$

where $z \in \mathbb{R}^d$ and the original ELBO is structured as:

$$\mathcal{L}(\theta, \phi; x) = E_{q_\phi(z|x)}[log \, p_\theta(x|z)] - I(x, z)$$

$$- D_{KL}\left(q(z)||\prod_{j=1} q(z_j)\right) - \sum_d D_{KL}(q_\phi(z_j)||p(z_j)) \tag{10}$$

Not like $\beta$-VAE, FactorVAE(Kim & Mnih, 2018) and $\beta$-TCVAE(Chen et al., 2018) introduce the hyperparameter in front of the first term in Eq. 9 which is referred to as the *total correlation* (TC) (Chen et al., 2018). By amplifying the penalty on this term, the dependence between the variables is reduced hence emphasising the disentanglement. The second term in Eq. 9 is defined as *dimension-wise KL divergence*, which puts constraints on the generated latent code $z$ and pushes them towards their predefined Gaussian prior (Li et al., 2020).

The total correlation is intractable since the prior contains mixtures with a large number of components. However, FactorVAE introduces a discriminator to approximate the the density ratio that arises in the TC term. In the training session, the model extract representations of two batch of input and then send them to the discriminator. The second batch of extracted representation will be randomly permuted and the goal for the discriminator is to identify whether its input is from $q(z)$ or not. The discriminator and the VAE are trained jointly for getting accurate estimation.

## A.3 THE IMPLEMENTATION DETAIL OF OUR PROPOSED METHOD

Table 3: The detailed structure of our proposed method

| Dataset | Modules | Parameters | | | |
|---|---|---|---|---|---|
| | | Layer | in_channel | out_channel | kernel_size |
| | ImageEncoder | Conv2d | 2 | 4 | 4x4 |
| | | Conv2d | 4 | 8 | 4x4 |
| | | Conv2d | 8 | 16 | 4x4 |
| | Feature Separation Module | Conv2d | 6 | 6 | 3x3 |
| | | Temporal Gated-Conv | 6 | 6 | 3 |
| BikeNYC/ TaxiNYC | | Conv2d | 6 | 6 | 3x3 |
| | | Conv2d | 6 | 6 | 3x3 |
| | Decoder | ConvTranspose2d | 8 | 16 | 4x4 |
| | | ConvTranspose2d | 4 | 8 | 4x4 |
| | | ConvTranspose2d | 2 | 4 | 4x4 |
| | Discriminator | Linear | 256 | 500 | ReLU |
| | | Linear | 500 | 500 | ReLU |
| | | Linear | 500 | 500 | ReLU |
| | | Linear | 500 | 2 | ReLU |

## A.4 EXAMINING THE LEARNT REPRESENTATIONS

**Quantitative evaluation of the learnt representations:** We summarise the results in Table 4. As one can see, our approach consistently shows superior performance across all datasets in both the

disentanglement and reconstruction. This verifies the necessity of extracting time-irrelevant features for each time step, which can capture the unconventional changes in the ST raster data very well.

Table 4: Quantitative performance comparison on BikeNYC, TaxiNYC and TaxiBJ datasets.

|  | BikeNYC | TaxiNYC | TaxiBJ |
|---|---|---|---|
| MMD | 0.858 | 0.918 | 0.354 |
| KLD | 0.108 | 0.131 | 8.305 |

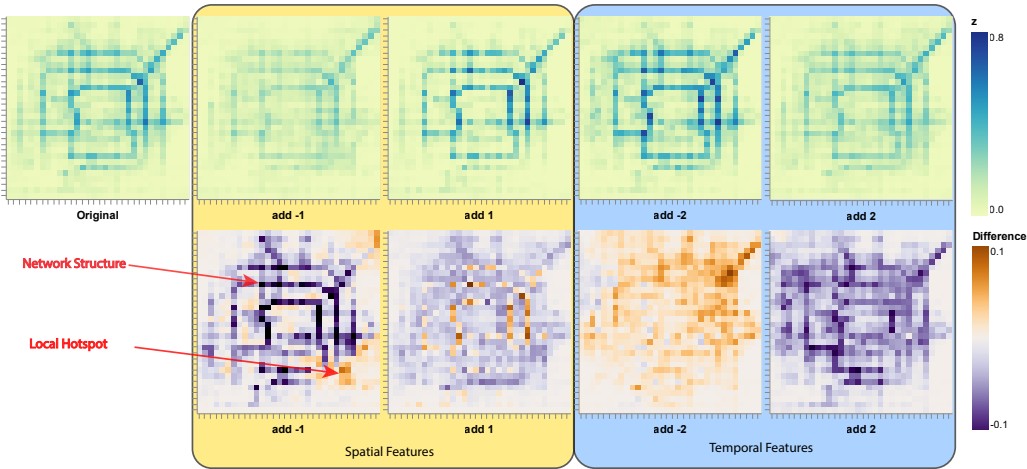

Figure 5: Spatiotemporal data generation controlled by fixing the spatial/temporal variable and modifying the other variable set. The top row shows the synthesis image with changed features, while the images on the second row show the difference between the original image and their corresponding synthetic images.

**Generating Attributed Spatio-temporal data:** We visualise the spatio-temporal data generated by our method to validate what types of information are extracted by the spatial/temporal features. The generated images are shown in Figure 5, and the original input is placed on the top left corner. The remaining of the top row shows the reconstructed spatiotemporal data with one group of features fixed (spatial/temporal) while making changes to the other set of variables. The difference between images on the first row and the original input is calculated and visualised in the second row since those residual images show the impact on the reconstructed data when changing the latent representations.

The temporal features, as shown in the right half of Figure 5, do not focus on the spatial structure of the data. They capture the temporal influences that flow through the sequence. When we are adding negative/positive values to temporal features, the model tends to amplify/weaken the temporal dependencies as we can see that the areas with non-zero values on the previous timestep are enhanced/reduced on this timestep. On the contrary, the spatial features contain more than the static structure of the transportation network. As shown in the left half of Figure 5, when adding/subtracting values to the spatial features, not only the flow of the major roads are amplified/weakened, some hotspot events also emerge across the whole city. The results above demonstrate that verify the necessity of the disentangling design of our approach, which can capture and separate the spatial and temporal features very well.

## A.5    REPRESENTATION SWAPPING

Besides the temporal, we also perform the representation swapping to show the ability of our proposed method to generate synthetic spatiotemporal data. Suppose two real sequences are given for spatial information and temporal information, denoted as $X_s$ and $X_t$. The synthetic spatiotemporal data are expected to preserve the spatial structure in $X_s$ and the temporal dependencies in $X_t$. The qualitative comparisons are shown in Figure 6

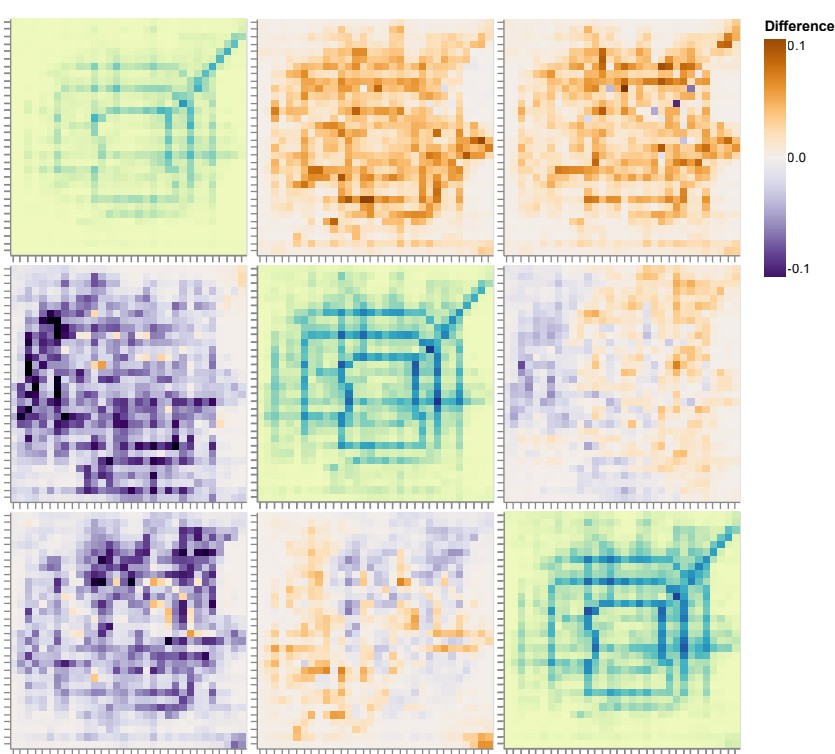

Figure 6: Spatiotemporal data generation controlled by swapping the spatail/temporal features

