# OpenReview forum: "Spatio-temporal Disentangled representation learning for mobility prediction"
_ICLR.cc/2022/Conference — ICLR 2022 Submitted_

### Official Review · Reviewer_MSHW · 2021-11-01

**Correctness:** 3
**Technical Novelty And Significance:** 2
**Empirical Novelty And Significance:** 2
**Recommendation:** 3
**Confidence:** 5

**Main Review:**

The authors approach an interesting and relevant problem such as the one of mobility forecasting. Specifically, the authors focus on how to leverage the disentanglement of spatial and temporal variability in order to get improved forecasting accuracy. I have however some concerns, which I would appreciate if the authors could help me clarify:

Introduction

The authors highlight two major challenges in literature: (i) “Difficulty in characterizing dynamic and spatial components”, (ii) “Difficulty in separating the extraction of spatial and temporal features”. Since the authors mention a variety of different works focusing on spatio-temporal forecasting, these claims could benefit from a more concrete narration (e.g. clarifying the meaning of “characterizing the dynamic and spatial components”). The reason being is that this part of the paper is responsible for motivating the additional complexity proposed by the authors. In other words, gaps in the literature, and the ways in which this work aims to fill these gaps, should be clear and tangible. Currently, I feel like I am failing to grasp what the authors mean here.

Section 3

- The authors explain rather quickly the data representation. I feel like a more detailed explanation of how flow data is generated from raw observations of mobility could improve the general readability of this paper. This might be obvious in transportation literature, but not necessarily for a machine-learning audience
- When defining the priors, the distributional form of $p(z_t^{Te})$ is not given, i.e. what is p(.) in $p(z_t | z_{<t})$?
- Eq (1):  (i) when defining the $\prod_{t=1}^T$, it seems like $p(z_t)$ depends on data from the full sequence (1:T), thus also using data from the future. Is this what the authors meant?
(ii) The authors define $p(z_{1:t}) = \prod p(z_t | z_{<t})$, with $p(z_t | z_{<t})$ showing a dependency on all previous ts. This is in apparent contrast with Figure 1. If I understand correctly, either (as in Figure 1) the authors assume a Markov property of the latent variables (i.e., z_t summarizes all info up to time t) or the Figure should be adjusted to reflect explicit dependency on all previous timesteps.
On the same line, if I am not mistaken, Eq (2) is also not consistent with Eq (1): the role of the product term in the two factorizations is different. In order to be consistent with Eq(1), I guess the authors meant $p(joint) = \prod_{t=1}^T p(x_t | z_t)p(z_t) = \prod_{t=1}^T p(x_t | z_t) p(z^{Sp}) p (z^{Te}) = p(x | z) p(z^{Sp}) \prod_{t=1}^T p(z_t^{Te} | z_{<t}^{Te})$.
- Eq (3): are the authors actually referring to $z_T$ or should it be $z_{1:T}$? From the factorization used later (product over T of $q(z_t^{Sp} | x_t)$ ) it seems more like the second option.
- Some background in variational inference could really help for readability (especially how beta-VAEs and FactorVAEs are placed w.r.t. literature, as these seem to be closely related to this work)
- The authors define the loss as $L_{vae} + L_{tc}$, and say "we estimate the total correlation using the same approach like FactorVAE". For clarity of exposition, I would explicitly write how L_tc is computed in the manuscript (authors could consider a background section to group all these key concepts from literature)

Experiments
- My main concern lies in (i) the misalignment between the experiments conducted and the contributions of the paper, and (ii) the relevance of quantitative results. Authors claim the architecture enables *disentanglement*, but no experiment actually tries to show/validate these claims. Rather, experiments focus on high prediction accuracy. In my view, high accuracy does not prove the better ability of the model to learn disentangled representation, but rather that the specific architecture can achieve better predictive performance
- Also, the results in Table 1 seem to be somewhat weak. Specifically, authors claim to be SOTA on two out of three, but, if I correctly interpret the Table,  on the TaxiNYC dataset, performance varies depending on which metric is considered (best in MAE, but 5th in RMSE). Also, it can be argued that on TaxiBJ results are not that comparable with the best performing model (having access to means and std. dev over repeated runs would help here)
- In general, I find the description of the experimental setup could be improved.

Moreover:
- Is the feature separation model pre-trained and frozen during the training of the other modules in the architecture? Also, the authors say they train the feature separation module on all data, could this have some (unwanted) spill-over of information from future observations? I feel the paper would gain from further elaborating on these points.
- In 4.2, I am not sure I understand what the disentangle modules are, is it different instantiations of the feature separation module described in section 3?
- Is the model using only spatial (or temporal) features retrained end-to-end (i.e. is the feature extractor also retrained?)
- Table 2: are means and std. dev. computed on different hyper-parameter settings? If this were to be the case, I find the statistic to be somewhat uninformative, as the different settings of hyper-parameters represent essentially different models. I would find it more informative if we were averaging over multiple runs of the same configuration. Moreover, if the variance for different hyper-parameters is so high, the authors should elaborate on how to select the best hyper-parameters values.
- What are the different configurations of hyper-parameters? I don’t seem to find this in the paper.
- I find the section on informative features very interesting. However, as in the comment above, I don't find the ablation over (unspecified) hyper-parameters very meaningful. For the purpose of exposition, I would probably select the best performing configuration and use that to showcase properties of the model

In conclusion, here is a list of typos and suggestions that can hopefully be useful:
- The authors should define terms before using them: ST, ST-raster, "Closeness, period, trend scheme”
- End of Section 3.2: "each frame xt of the by using"
- Eq(5) the authors can probably unclutter the equation by aggregating summation terms between the two KL

**Summary Of The Paper:**

The authors propose a novel neural architecture to structurally learn disentangled spatio-temporal representations in the context of mobility forecasting. This work argues that predictive models can benefit from enforcing the independence of spatial and temporal dynamics and proposes a VAE-inspired architecture for doing so.
By conducting experiments on three mobility datasets, the authors empirically evaluate the prediction performance of the proposed approach against various baselines. Additionally, further experiments attempt to (i) quantify the effectiveness of different temporal features (short-term/daily/weekly correlation), and (ii) formalize and justify a strategy for feature selection.

**Summary Of The Review:**

The authors approach an interesting and relevant problem and propose an interesting avenue for improving mobility forecasting techniques. However, I find the main issue of this paper to be the weakness of the empirical results, making the contribution, in my opinion, rather marginal. The methodological section can also be improved, as it leaves many open questions to the reader on a number of design choices. Because of this, I believe the current version of the paper is not yet worthy of publication.

In my opinion, in order to improve the current version of the manuscript, I believe the authors should focus on:
- Additional experiments to justify/prove the claims on “spatio-temporal disentanglement”
- Clarify experimental design (e.g. settings of hyper-parameters)
- Clarify the significance of results in Table 1
- Improve/Revise the presentation of the math describing the proposed architecture (i.e., Section 3)

---

> ### Author Response · Authors · 2021-11-22
> **Thank you for the repsonse.**
>
> Q1. Introduction:
> Thank you for your comment. In the introduction, we wrote about the difficulty in characterising the dynamic and spatial components since the current state-of-the-art methods claimed that they can better extract long-range spatial and temporal features, but they failed to demonstrate why their representation or model is better except for better prediction results. We find some interesting results about characterising the dynamic and spatial components when we perform the feature traversal on our learned representation (in appendix A.4 and A.5). However, we struggle to visualise the results since all three datasets that we used in this work are datasets without semantic labels. However, the current introduction part does need refinement to align the motivation and the contribution.
>
> Q2. Section 3:
> Thanks for this comment. To this end, we revised both section 3 and the appendix to include more details about the equations.
>
> Q3. Experiments:
> To this end, we revised the paper to add experiments using our methods without explicit enforcing the disentanglement, and the results are shown in Table 1. It shows that without the explicit constraints on the disentanglement, our model deteriorates to the stack of several CNN and a normal VAE component, and its results on all three datasets are just slightly better than the CNN baseline. This result justifies the explicit constraints on the disentanglement.
> As for the quantitative evaluation, we include a table in the appendix containing the MMD and KLD scores of our learned representations. There are few unsupervised evaluation metrics for the level of disentanglement. We are trying to add UDR scores, but it takes a considerable amount of time to run.
>
> Q4. Is the feature separation model pre-trained and frozen during the training of the other modules in the architecture?
> The feature separation module is trained on all data before we use it for the mobility prediction task. Therefore, when we use it to make predictions, the feature separation module is frozen, and only the MLP predictor is trained to make the next-step prediction. The reason is that when we train the feature separation module, the objective function does not contain anything that could be relevant to the downstream task, and it will only be used to reconstruct the original sequence. Therefore, it should not contain unwanted knowledge for future observations.
>
> Q5. In 4.2, I am not sure I understand what the disentangle modules are. Is it different instantiations of the feature separation module described in section 3?
> The disentangle modules used in section 4.2 are the feature separation module described in section 3.
>
> Q6. Is the model using only spatial (or temporal) features retrained end-to-end (i.e. is the feature extractor also retrained?)
> No, we added masks on either the spatial (or temporal) features since we want to investigate their contribution to the mobility prediction problem.
>
> Q7.  Table 2
> No, the means and standard deviation are computed across all the different representations learned with different random seeds and hyperparameters. As discussed in section 4.4, the quality of the learned representations heavily depends on random seed and hyperparameter setup. Since we want to make sure these strange results we get (the configuration c6 wasn't the best setup) and eliminate the impact for a particular seed or hyperparameter, we trained multiple feature separation modules with different seed and hyperparameters and calculated the means and standard deviation on all those setups.
> The different combinations of the hyper-parameters contain the depth of the Image encoder, the number of spatial/temporal features learned, the learning rates and whether to use the instance normalisation layer.
> The results show that the quality does vary significantly for different random seed and hyperparameter setups, but in general, configuration c6 was not the best configuration for all three datasets.

---

### Official Review · Reviewer_z9DA · 2021-11-01

**Correctness:** 3
**Technical Novelty And Significance:** 2
**Empirical Novelty And Significance:** Not applicable
**Recommendation:** 3
**Confidence:** 5

**Main Review:**

Strengths:

1. Spatio-temporal representation learning is an important problem.
2. The proposed solution is evaluated on several real-world datasets.

Weaknesses:

1. The biggest concern for this paper is the motivation. Why is it a good idea to disentangle spatial and temporal representations? This idea does not seem to make sense and is not sufficiently justified in the paper. Typically, the interactions between the spatial and temporal dimensions (i.e., spatio-temporal autocorrelation) is the most challenging component to model in spatio-temporal data. For example, the dynamically changing footprint of traffic congestion on a road network is a function of both location and time. Forcing the separation of the spatial and temporal dimension might cause such dependencies to be ignored in the downstream analysis steps. The authors fail to provide a convincing reason to do so.

2. It appears that using generative models for spatio-temporal data modeling has been studied previously. The authors may consider checking the following works that use conditional GAN models for traffic estimation:

[1]. Zhang et al. Curb-GAN: Conditional Urban Traffic Estimation through Spatio-Temporal Generative Adversarial Networks. In KDD 2020.
[2]. Zhang et al. TrafficGAN: Off-Deployment Traffic Estimation with Traffic Generative Adversarial Networks. In ICDM 2019.

While VAE is a different generative model, it is very relevant to the above works. The authors may want to demonstrate the differences between their work and these existing methods. They should also be compared in the evaluation section as baselines.

**Summary Of The Paper:**

This paper proposes a variation of the Variational Autoencoder (VAE) model to learn disentangled spatial and temporal representations from ST raster data. A separation module is designed in the proposed network. The proposed model is validated on three real-world datasets and achieves better performance compared with the baselines listed in the paper.

**Summary Of The Review:**

Overall, the paper studies a topic in an important domain. However, the motivation of the work is questionable and unjustified. It does not seem to be beneficial to disentangle spatial and temporal information from an ST dataset. The idea of using generative models for ST data modeling has been explored before, which was ignored by the authors.

---

> ### Author Response · Authors · 2021-11-22
> **Thank you for the response.**
>
> Q1. Why is it a good idea to disentangle spatial and temporal representations?
> Thanks for your comments. To this end, we revised the paper to add experiments using our methods without explicit enforcing the disentanglement, and the results are shown in Table 1. It shows that without the explicit constraints on the disentanglement, our model deteriorates to the stack of several CNN and a normal VAE component, and its results on all three datasets are just slightly better than the CNN baseline. This result justifies the explicit constraints on the disentanglement.
> We also find some interesting results when we perform the feature traversal on our learned representation (in appendix A.4). However, we struggle to visualise the results since all three datasets that we used in this work are datasets without semantic labels. We made some illustrations in appendix A.4 and A.5, but I think it indeed is not enough to justify our choice when designing our framework.
>
> Q2. More baseline on the spatio-temporal data modelling:
> Thank you for this comment. The two existing works mentioned by the reviewer uses generative models for spatio-temporal data modelling that achieve promising results. However, they focus on generating the graph-based traffic data by learning the network structure instead of the raster-based data used in this work. We try to disentangle the extract features better to gain more insight into the direction of further improvement.

---

### Official Review · Reviewer_ZJKW · 2021-11-02

**Correctness:** 2
**Technical Novelty And Significance:** 2
**Empirical Novelty And Significance:** 2
**Recommendation:** 3
**Confidence:** 4

**Main Review:**

Strengths:
- This paper is generally easy to read and follow.
- The applications considered in this paper are quite relevant and can have a high real-world impact.

Weaknesses:
- The level of novelty is rather low. This paper builds heavily on ideas from the references (Duan et al., 2019) and (Kim & Mnih, 2018). The authors should make it very clear how this paper differs from the state of the art, and particularly those two references.
- The motivation for the need for disentangled representations is not clear, nor is there strong empirical evidence suggesting that need. The authors should consider improving the justification of why are disentangled representations necessary for mobility prediction problems. Also, they should consider including empirical evidence of how disentangled representations for spatial and temporal features improve predictive performance - an ablation study could be used for this.
- The explanation of the methodology is not thorough enough, and a lot of key details are missing from the paper. This hurts readability and reproducibility. For example, how is the sequential prior z_{1:t} = z_t | z_{<t} defined?  In Eq. 6, how is p_\theta (x_t | z_t^Sp, z_t^Te) defined? How is the total correlation used in the VAE loss defined?
"In this work, we estimate the total correlation using the same approach like FactorVAE (Kim & Mnih, 2018), i.e. by introducing a discriminator and using the independence testing trick and the density-ratio trick to approximate the KL term in the above equation." -> Can you provide additional details? For example, how is the discriminator defined?
- The paper does not account, nor refer, to important recent works on modelling spatio-temporal data. Especially, on GNN-based methods, which have recently shown very promising results. A few examples:
"Zonghan Wu, Shirui Pan, Guodong Long, Jing Jiang, Xiaojun Chang, and Chengqi Zhang. Connecting the dots: Multivariate time series forecasting with graph neural networks. In Proceedings of the 26th ACM SIGKDD International Conference on Knowledge Discovery & Data Mining, pp. 753–763, 2020."
"Zheng Fang, Qingqing Long, Guojie Song, and Kunqing Xie. Spatial-temporal graph ode networks for traffic flow forecasting. In Proceedings of the 27th ACM SIGKDD Conference on Knowledge Discovery & Data Mining, pp. 364–373, 2021."
"Yuzhou Chen, Ignacio Segovia-Dominguez, and Yulia R Gel. Z-gcnets: Time zigzags at graph convolutional networks for time series forecasting. International Conference on Machine Learning, 2021."
The authors should also consider including some of these as baselines.
- A key baseline is missing: the proposed model without disentanglement. The authors should consider including an ablation study on several key components of their contribution such as total correlation in the VAE loss. This could allow them to justify the need for disentangled representations.
- The improvements over naive baselines such as a simple ConvLSTM seem quite small. Are these improvements statistically significant? Have the authors considered performing multiple runs and reporting averages, standard deviations and performing statistical significance tests?

**Summary Of The Paper:**

This paper proposes learning a disentangled representation of spatio-temporal mobility data using a VAE-based architecture, which essentially tried to decompose spatial and temporal features and model them independently. Disentanglement in the learnt representation is encouraged through the introduction of total correction as a regularization term in the VAE loss. The authors empirically demonstrate the potential of the proposed approach in 3 different mobility datasets (BikeNYC, TaxiNYC and TaxiBJ). Overall, the results suggest competitive predictive performance with other state-of-the-art approaches.

**Summary Of The Review:**

In summary, this paper is interesting to read and easy to follow, but the work needs to be improved before being ready for publication. There are several key issues related to the presentation (e.g. clarity and level of detail of explanations), experimentation, empirical validation of several important claims, state-of-the-art references, and baselines, that should be addressed. Unfortunately, the level of novelty is also rather low.

---

> ### Author Response · Authors · 2021-11-22
> **Thank you for the response!**
>
> Q1. Novelty:
> Thank you for your comments. The differences between our methods and the current state-of-the-art methods are discussed in Section 3.2. The existing methods assume that the factors comprise two different parts: one dynamic feature to capture the temporal dependencies and a set of fixed content features shared by the whole sequence. It works for the simple sequence datasets like Moving MNIST and Sprite because the content in those sequences does change throughout the whole sequence (usually an Avatar or some numbers). However, for the spatio-temporal data, this fixed content feature set is not enough. To address this gap, we proposed our method where each timestep has its own spatial and temporal features.
> As for the differences between our method and prior works (Duan et al., 2019) and (Kim & Mnih, 2018), the standard factorVAE(Kim & Mnih, 2018) was proposed for image disentanglement. Therefore it does not suit spatio-temporal data. In our work, we introduced the discriminator in factorVAE to estimate the Total Correlation regularised. Prior work (Duan et al., 2019) focuses on unsupervised model selection (so it didn't propose any new approaches in terms of better disentanglement). We just introduce its standard to distinguish whether a learned feature is informative or not.
>
> Q2. More technical details:
> To this end, we revised the paper and added the more detailed definitions in the appendix (due to the page limit).
>
> Q3. GNN-based methods:
> Thank you for this comment. We do aware that there are some existing works that show promising results using Graph-based methods. However, the reason we did not include them as reference is that the dataset used by those methods are graph-based datasets (California traffic speed dataset and etc,). Therefore, it isn't easy to apply them as baselines on the datasets that we used in this work. Besides, our method also cannot be directly applied to graph-based datasets since it is designed for spatio-temporal raster data.
>
> Q4. Baseline (the proposed model without disentanglement):
> To this end, we have already revised the paper and added the results for our method without explicit disentanglement in Table 1.
>
> Q5. The improvements over naive baselines are not significant:
> We showed the results of our proposed methods in the second part of Table 1, and it achieved the best results for the dataset TaxiNYC and BikeNYC. We did run multiple experiments for different seeds and hyperparameters, and the results are summarised in Table 2. It shows that our methods can achieve better results with different seed and hyperparameter setups (although the variance is a bit large due to the impact of different hyperparameter setups).

---

### Official Review · Reviewer_f7db · 2021-11-04

**Correctness:** 4
**Technical Novelty And Significance:** 3
**Empirical Novelty And Significance:** 2
**Recommendation:** 5
**Confidence:** 3

**Main Review:**

Pros:
* The core idea of the paper, that spatial features are not mandatory static and have to be generated at each timestep is a very interesting research direction in the forecasting setup.
* The approach is well motivated compared to SOTA
* The experiments are well-conducted and supplementary materials give useful insights into the learned representations.
* The philosophy of the model is well described and the overall architecture is comprehensible.

Cons:
* There are many technical details that are missing to be able to reproduce the architecture and the experiments. The different blocks of the architecture are not detailed (neither in the appendixes), there is no indication for instances on the number of layers, size of the latent spaces, and so on.
* The description of the model relies too much on references (gated conv. unit, Total correlation are not introduced).
* A curious aspect of the reported results is not analyzed:  the ablation study shows that the configuration  C0 is clearly the best one, which includes only the closeness without period and trend. What is the point to present the configuration C6 as the main result in this case? And what is the meaning of this phenomenon? The extracted features from the past timestep are sufficient to achieve the best forecasting without taking into account long-range dependencies or seasonality ? It is a very strange result.


**Summary Of The Paper:**

The paper proposes a VAE model for mobility forecasting. It focuses mainly on the disentanglement of spatial and temporal features by modelizing explicitly two groups of features in the VAE schema, a group of spatial features that are time-independent and a group of temporal features using a sequential prior. In order to make a prediction, the VAE is used to extract features from three groups of sequences describing the trend, the period, and the last few timesteps before the forecasting period.  An ablation study is provided concerning those three groups of sequences. Experiments show competitive results wrt SOTA.

**Summary Of The Review:**

The core idea of the paper is very interesting for the problem of mobility forecasting but as it is the description of the model is too broad to reproduce the architecture and the experiments.  Results show some surprising effects that are not analyzed.

---

> ### Author Response · Authors · 2021-11-22
> **Thank you for the response.**
>
> Q1. More technical details:
> To this end, we revised the paper and added more technical details about our proposed framework (you can find them in the appendix).
>
> Q2. Why use the configuration C6 when C0 is the best:
> Thank you for your comments on this aspect. The reason we introduce the "closeness, period and trend" scheme in our experiments is that other baseline models (the current state-of-the-art methods in the mobility prediction area) use this setup. And in order to have the foundation to compare our results with theirs, we first run the experiment under this setup with our proposed method (Table 1). After that, we want to see which component contribute most to the result since all recent works focus on extracting the long-range temporal dependencies to get better prediction results. And surprisingly, we got the results that were shown in Table 2. However, we didn't have enough time to investigate why we got such results when we submitted this work. This could be a good future direction.

---

### Decision · Program_Chairs · 2022-01-20

**Decision:**

Reject

**Comment:**

The paper proposed to learn a disentangled representation of spatiotemporal mobility data using a VAE-based architecture, in order to separate spatial and temporal dependencies. This is an interesting and relevant problem, but the reviewers found the paper to be weak in motivation and empirical evaluations.